# Design Analysis of Prosthetic Unilateral Transtibial Lower Limb with Gait Coordination

Susmita Das [1,2,*], Dalia Nandi [2] and Biswarup Neogi [3]

1   Electronics and Instrumentation Engineering Department, Narula Institute of Technology,
    Kolkata 700109, West Bengal, India
2   Electronics & Communication Engineering Department, Indian Institute of Information Technology,
    Kalyani 741235, Nadia, India; dalia@iiitkalyani.ac.in
3   JIS College of Engineering, Electronics and Communication Engineering,
    Kalyani 741235, Nadia, India; biswarupneogi@gmail.com
*   Correspondence: susmitad2011@gmail.com; Tel.: +91-8017205403

**Abstract:** People with lower limb amputations struggle through difficulties during locomotion in their daily activities. People with transtibial amputations take support from prosthetic legs for systematic movement. During motion, they experience some mobility issues while using general prosthetic limbs regarding gait pattern. The design of a prosthetic-controlled lower limb with gait synchronization for physically disabled persons is the main goal of the present research work, which can provide an improved walking experience. The design and performance analysis of prosthetic lower limbs for people with transtibial amputations is performed in the present paper. The designed rehabilitation system shows synchronization between the normal and the prosthetic limbs achieved with gait coordination. The dynamics of the lower extremities in different postural activities are used for design purpose utilizing Euler–Lagrange motion theory. The artificial motion of the knee and the ankle joints function through the angular movement of the servo motors according to the movements of the rotary encoders placed on the sound limb joints. The range of motion of both the sound and prosthetic limbs are compared for different steps during a gait cycle. The prosthetic electronic system design of the artificial lower limb is able to show the gait style of human being with body kinesics. The nonlinear domain stability analysis of the designed prosthetic limb is presented through the Lyapunov method. A PIDF2 controller tuning process is implemented for the designed limb's performance improvement. The designed prosthetic system is beneficial for people with unilateral transtibial amputations with a great societal impact.

**Keywords:** people with transtibial amputations; prosthetic limb; servo motor; gait coordination; nonlinear control

## 1. Introduction

People who have undergone amputations and have one normal lower limb can control the synchronized movement of a prosthetic leg attached to the residual part of the amputated limb to experience normal walking pattern. According to the range of motion of a normal leg, an artificial leg moves as per the prosthetic arrangement to provide a humanoid stride for motion. The hardware design of the artificial system includes the movement signal acquired from the existing limb of a person with an amputation to replicate the leg movement. The total constructional design works as a humanoid robotic leg with a prosthetic attachment and rehabilitation features for people with transtibial amputations. The full setup is controlled by networking and programming logic to obtain a balanced and symmetrical walking style. A person with a unilateral transtibial amputation can retrieve their previous capability to walk with a rehabilitation process. A controller-based exoskeleton design operation is described for the clinical outcome for people who have undergone amputation [1]. Single actuator-powered lower limb prosthesis design with

kinematics is described in [2]. An elaborate review work is presented regarding system design of lower extremity prosthetic system in [3]. Recent advancements to lower limb exoskeleton design are considered in [4]. The mechanical human motion with functionality is described in [5]. The robotic motion planning, control and mechanics have been described in [6]. Lower limb joint prosthesis-oriented design has been shown in [7]. Rehabilitation related to the bounded control method for joint movements has been presented in [8]. A power-assisted active interface has been presented in [9]. Artificial human ankle movement has been implemented in [10]. A review of lower limb rehabilitation design has been presented in [11]. Leg motion experiments with knee flexion analysis have been performed in [12]. Joint rehabilitation-oriented lower limb design has been implemented in [13,14]. Human movement data acquisition was performed in [15,16]. In the recent research, prosthetic system design for people with transtibial amputations has been presented with mechanical structure [17,18]. The most unexplored part is electronic system design for gait synchronization between the normal limb and the prosthetic limb with body gesture. Until now, mechanical exoskeletons and prosthetic models have been designed for people with lower limb amputations to be capable of performing the daily tasks. The gait synchronization of the artificial limb with and the healthy limb utilizing kinesiological analysis is the goal of the current work alongwith the natural body movement. According to the gait pattern of the normal limb of a person with an amputation, the artificial limb's movement is the main unexplored job for forward movement process. The Range of Motion (RoM) implementation with proper functionality for the unilateral transtibial amputated leg is a matter that has not been executed in previous research works until now. The theory of Lyapunov stability is a standard theory for non-linear systems and one of the most important mathematical tools in the analysis of non-linear systems in robotic design [19] which has been applied in the present work.

## 2. Methodology

Building a controlled prosthetic lower limb with gait synchronization for people who have undergone amputation is the most challenging job. Human factors related to knowledge are reflected through this present work [20]. The lower extremity solution gives artificial movement of the residual leg of an amputated person according to the gait style of the existing or normal leg. This method depicts the function of prosthetic model for artificial lower limbs with the joint movement signals of human legs.

### 2.1. Technological System Design

In Figure 1, the schematic diagram of a gait-synchronized artificial limb design process is presented [21]. Initially, the output voltages of the joints (knee and ankle) in the normal leg were acquired using rotary encoders. The rotary encoders were attached at the knee and ankle joints to observe the angular movement value during motion, as shown in Figure 2. The data were recorded for different muscular activities related to lower limb movement for a complete gait cycle [22]. Motor movements for an artificial limb were processed using the different angular movements during gait, replicating those of the normal leg. Mobility with synchronization in walking patterns is a novelty of this work.

Rotary encoders of the normal right limb knee joint and ankle joint were denoted as $R.E_{knee}$ and $R.E_{ankle}$, respectively. Servo motors of the prosthetic left limb knee joint and ankle joint were denoted as $M_{knee}$ and $M_{ankle}$. The normal right limb of people who have undergone amputation as the transmitter was denoted as $NL_{Tx}$ and the prosthetic left limb of a person with an amputation as the receiver was denoted as $PL_{Rx}$. In the present study, goniometers are used for measuring the angular movements at ankle and knee joints of normal and prosthetic limbs.

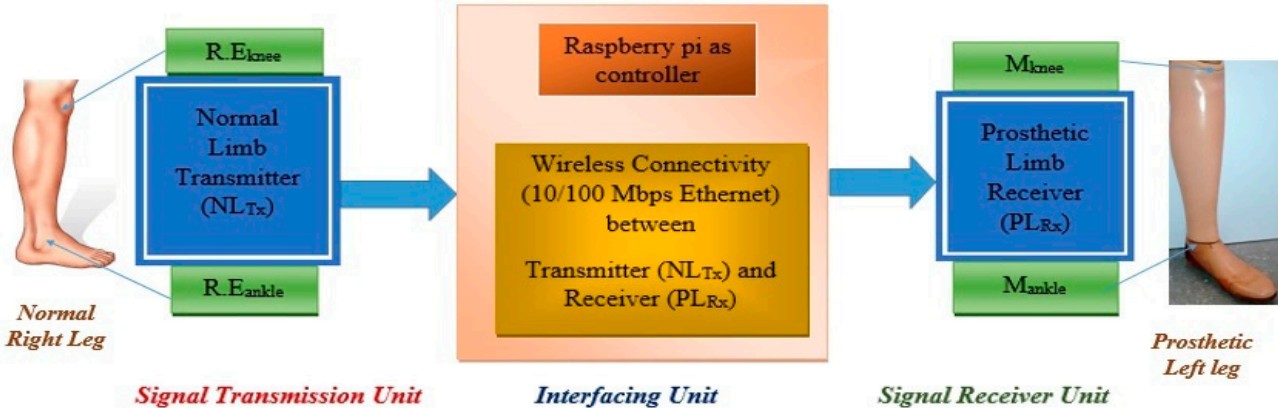

**Figure 1.** Schematic Design of Gait-Synchronized Prosthetic Limb with Normal Limb.

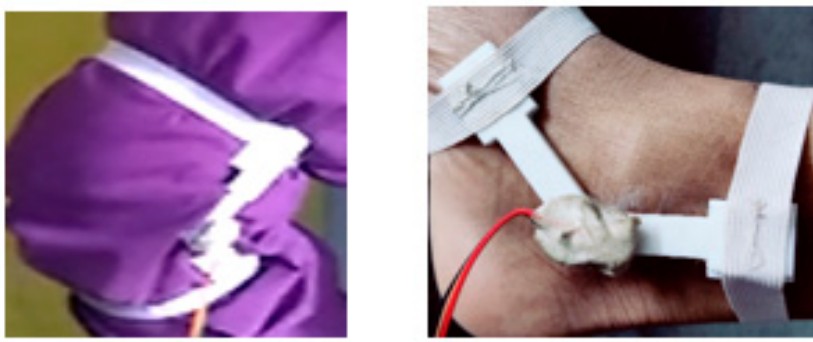

**Figure 2.** Pictorial Representation of Rotary Encoder connected at knee and ankle joints.

### 2.2. Embedded System Implementation of an Artificial Lower Limb

The artificial leg model is considered to represent the design of the electronic prosthetic system presented in Figure 1. The joint movement signal was given through rotary encoders attached at the knee and ankle joints of a normal limb with a predefined range of motion to be implemented on the artificial model [23,24]. Here, according to the existing limb movement, the artificial leg moved with proper gait style in the prosthetic system design. Raspberry pi 3b+, Arduino Nano, Node MCU, 16 channel Servo driver (PCA 9685), Servo motor MG 995 were needed to implement the prosthetic system with portability. Raspberry pi 3b+ controlled the whole system as the main controller, and Arduino Nano was used to implement the movement of the limb as per the output of the rotary encoders. PCA 9685 drove the servo motors (MG 995) placed at two different joints of the leg model. These were used to perform the movements of the joints to help the amputee person to walk with gait synchronization. The wireless connection maintained the serial communication with a 9600 BAUD (Bits of Actual Usable Data) rate. The system was made portable using a 12 volt lithium ion battery with better power usage compared with the other types. This operated at a higher voltage than other rechargeable batteries with lower rates of charge loss.

The comparison between sound limb and artificial lower limb with respect to gait functionality is shown to present the prosthetic system's performance. The model is beneficial for people who have undergone amputation to experience a normal walking style. To solve the challenging part of facing any posture change in the progression, inverse kinematics was applied, which is explained in detail in Section 3.1.

### 2.3. Gait Synchronization Process

Stance and swing phases are two stages of a human gait cycle. The initial step where the contact of the foot starts from the ground, followed by consecutive steps, represents stance phase. In total, 60% of gait cycle includes stance zone and almost 40% of gait cycle involves swing phase. The swing phase shows the support of a single leg where the foot is

not connected to the ground. The most important seven steps to complete the gait cycle of a human being are heel strike, foot flat, midstance, push off, acceleration, mid-swing and deceleration. The initial four steps are included in the stance phase and the rest of the steps are involved in the swing phase. The movement angles with respect to the ground were measured with a goniometer to show Range of Motion (RoM) of normal limb and prosthetic limb of a person with a unilateral transtibial amputation. The RoM of the knee joint of the normal limb ranged from 0 degrees to 150 degrees for flexion and 120 degrees to 0 degrees for extension. The RoM of the ankle joint of the normal limb was from 0 degrees to 40 degrees for plantar flexion and 0 degrees to 20 degrees for dorsiflexion. The RoM of the knee joint of the designed prosthetic limb was from 120 to 180 degrees. The RoM of the ankle joint of the designed prosthetic limb was from 0 to 56 degrees [17]. In Figure 3, the pictorial representation of the normal and the designed prosthetic lower limb design are shown.

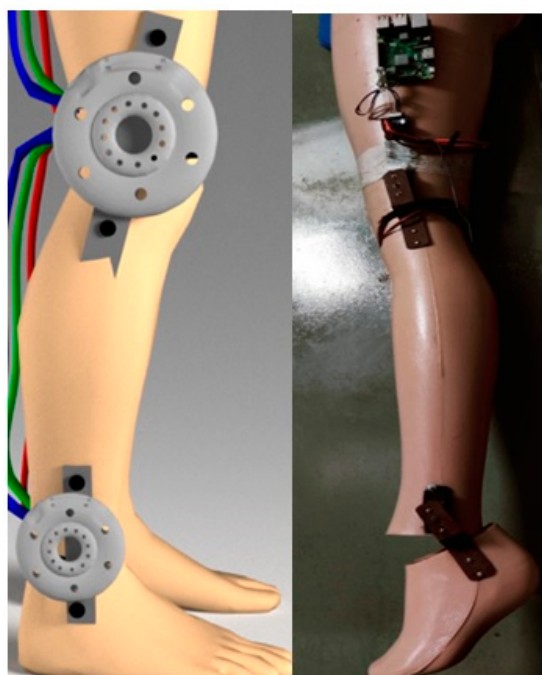

**Figure 3.** Pictorial Representation of Right Normal Limb and Left Prosthetic Limb.

### 3. Results and Discussion

The gait performance analysis with respect to the seven steps of the normal and the designed prosthetic limb is shown in Figures 4 and 5. The knee and ankle joint movements of the prosthetic limb show a balanced condition all the steps of a single gait cycle. In each and every step of one gait, the angular motion of the joints and the corresponding output voltage values were compared with the normal limb's movement to represent the differences in gait pattern for a body balancing feature creation using synchronization. In Table 1, the gait angular output variation of the normal and prosthetic limb are presented. The angular measurements are taken using Goniometer connected at the joints. It was observed that the prosthetic limb was capable of performing the synchronized gait pattern for people with transtibial amputations with a balanced condition of body kinesics.

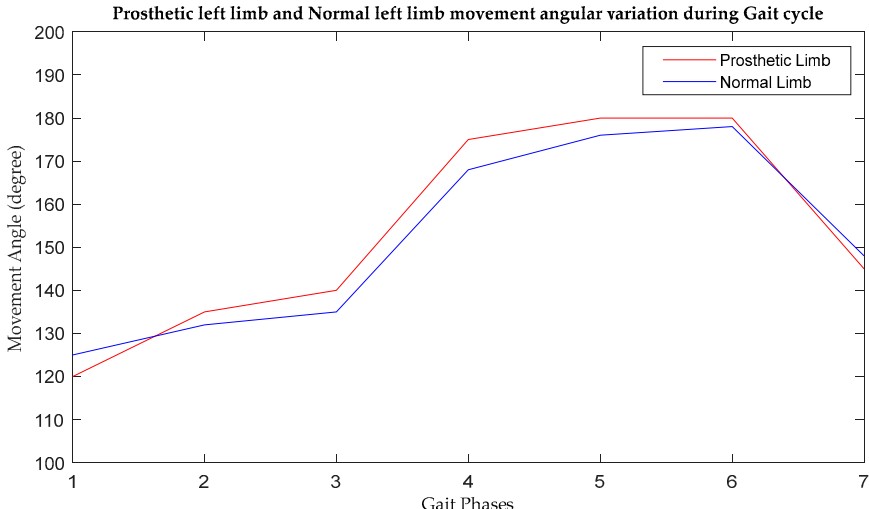

**Figure 4.** Graphical presentation of the Prosthetic left limb and Normal left limb movement of Knee joint during Gait cycle.

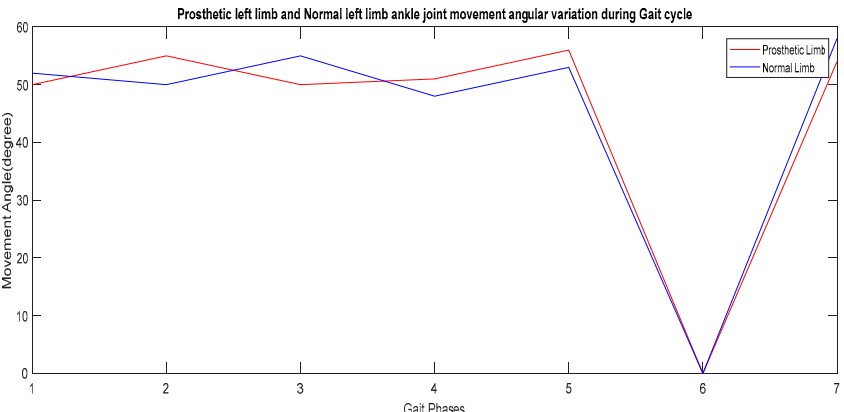

**Figure 5.** Graphical presentation of the Prosthetic left limb and Normal left limb movement of Ankle joint during Gait cycle.

**Table 1.** Gait angle variation in the normal right limb (rotary encoder), the prosthetic left limb (Servo motor) and the normal left limb.

| SL. No. | Gait Phases | Movement Angle of the Normal Right Limb (Degree) | | Movement Angle of the Prosthetic Left Limb (Degree) | | Movement Angle in the Normal Left Limb (Degree) | | Angular Deviation between Prosthetic Left Limb and Normal Left Limb (Degree) | |
|---|---|---|---|---|---|---|---|---|---|
| | | Knee Joint ($R.E_{knee}$) | Ankle Joint ($R.E_{ankle}$) | Knee Joint ($M_{knee}$) | Ankle Joint ($M_{ankle}$) | Knee Joint | Ankle Joint | Knee Joint | Ankle Joint |
| 1 | Heel strike | 180 | 20 | 120 | 50 | 125 | 52 | 5 | 2 |
| 2 | Foot flat | 170 | 0 | 135 | 55 | 132 | 50 | 3 | 5 |
| 3 | Midstance | 180 | 2 | 140 | 50 | 135 | 55 | 5 | 5 |
| 4 | Push off | 125 | 54 | 175 | 51 | 168 | 48 | 7 | 3 |
| 5 | Acceleration | 140 | 51 | 180 | 56 | 176 | 53 | 4 | 3 |
| 6 | Mid-swing | 125 | 50 | 180 | 0 | 178 | 0 | 2 | 0 |
| 7 | Deceleration | 180 | 18 | 145 | 54 | 148 | 58 | 3 | 4 |

From Figures 4 and 5 it has been clearly observed that the movement angle variations for knee and ankle joints of the prosthetic limb is closely followed by the normal limb

movement. The changes in the 6th gait phase due to ankle movement is justified as per feet position on the ground as shown in Figure 5.

In Table 2, the gait output voltage variation in the rotary encoder placed on the normal limb joints and the servo motors placed in the prosthetic limb are presented. The voltage variations obtained from the prosthetic limb is almost similar to that of the normal limb for all the gait phases. It is observed that the prosthetic limb is capable to perform the synchronized gait pattern for people with transtibial amputations with changed value of potential outcome due to changes in the angular measurement.

**Table 2.** Gait output voltage variation in the Normal right limb (rotary encoder), the prosthetic left limb (Servo motor) and the Normal left limb.

| Sl. No. | Gait Phases | Movement Output of the Normal Right Limb (Volt) | | Movement Output of the Prosthetic Left Limb (Volt) | | Movement Output in the Normal Left Limb (Volt) | | Output Deviation between Prosthetic Left Limb and Normal Left Limb (Volt) | |
|---|---|---|---|---|---|---|---|---|---|
| | | Knee Joint ($R.E_{knee}$) | Ankle Joint ($R.E_{ankle}$) | Knee Joint ($M_{knee}$) | Ankle Joint ($M_{ankle}$) | Knee Joint | Ankle Joint | Knee Joint | Ankle Joint |
| 1 | Heelstrike | 5.1 | 5.1 | 5.5 | 5.3 | 5.7 | 5.0 | 0.2 | 0.3 |
| 2 | Footflat | 5.0 | 5.2 | 5.1 | 5.1 | 5.0 | 5.5 | 0.1 | 0.4 |
| 3 | Midstance | 4.9 | 6.2 | 5.2 | 5.1 | 5.4 | 5.3 | 0.2 | 0.2 |
| 4 | Pushoff | 4.9 | 4.9 | 5.3 | 5.2 | 5.2 | 5.2 | 0.1 | 0.0 |
| 5 | Acceleration | 4.9 | 4.1 | 5.4 | 5.4 | 5.4 | 5.2 | 0.0 | 0.2 |
| 6 | Mid-swing | 4.8 | 4.2 | 5.6 | 5.2 | 5.3 | 5.3 | 0.3 | 0.1 |
| 7 | Deceleration | 4.9 | 4.2 | 5.5 | 5.3 | 5.4 | 5.3 | 0.3 | 0.2 |

The knee and ankle joint movements of the prosthetic limb show a balanced condition during walking or the steps in a single gait cycle. In each and every step, the angular motion and output voltage values were compared with the normal limb's movement to represent the difference. For all the gait phases, the voltage deviation was much less, such as below 0.5.

Mathematical model generation is required to present the behavioral characteristics of the designed system, as shown in Figure 6. The stability improvement of the designed lower limb prosthesis is essential for balanced walking.

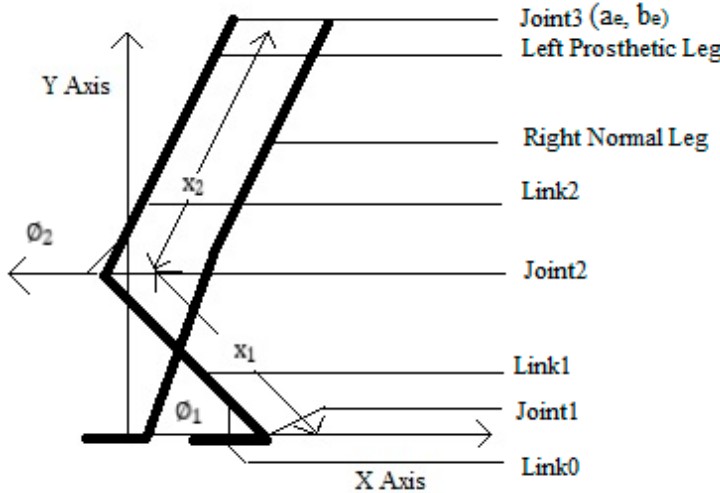

**Figure 6.** Schematic normal and prosthetic lower extremities with length and angular variations.

The postures of sitting and standing were observed for designing the most suitable system for people with lower limb activity challenges. Two specific cases of angular limb motions and lengths of upper and lower links of the lower extremity are shown in Table 3.

**Table 3.** Measurements of upper and lower links of a designed lower limb for two specific cases of limb motions with respect to the ground.

| SL. No. | Posture Condition of Limb | Length of Lower Link $x_1$ (Inches) | Length of Upper Link $x_2$ (Inches) | Angle of Lower Link $\varphi_1$ (Degrees) | Angle of Lower Link $\varphi_2$ (Degrees) |
|---------|---------------------------|-------------------------------------|-------------------------------------|------------------------------------------|------------------------------------------|
| 01 | Sitting | 16 | 18 | 52 | 22 |
| 02 | Standing | 16 | 18 | 87 | 89 |

*3.1. The Sitting Condition as Case I (17.5 Inches from the Floor)*

Measurements were acquired from the lower limb utilizing standard measuring apparatuses such as a ruler and a goniometer to measure the angular variation considering a schematic structure of a normal and prosthetic lower limb system, as shown in Figure 6. In this schematic diagram, a human lower limb is shown. Three joints and three links are presented in the diagram. The first (joint 1), second (joint 2) and third joints (joint 3) represent the ankle joint, the knee joint and the hip joint, respectively. The hip joint is connected to the upper half of the body. The first link (link 0) represents the feet, the second link (link 1) represents the lower half of the leg and the third link (link 2) represents the upper half of the leg. The coordinate [6] of the hip joint is $(a_e, b_e)$ and the joint displacements in angular forms are $(\varnothing_1, \varnothing_2)$, as given by the kinematic [7,8] Equations (1) and (2), as given below.

$$a_e(\varnothing_1, \varnothing_2) = x_1 \cos \varnothing_1 + x_2 \cos(\varnothing_1 + \varnothing_2) \tag{1}$$

$$b_e(\varnothing_1, \varnothing_2) = x_1 \sin \varnothing_1 + x_2 \sin(\varnothing_1 + \varnothing_2) \tag{2}$$

The acquired measurements of the designed lower limb are given below:

$x_1$ = lower link length between knee and ankle joints = 16 inches;
$x_2$ = upper link length between knee and hip joint = 18 inches;
$a_e$ = angle between the lower link and *x* axis for joint 1 = 52°;
$b_e$ = angle between the lower link and *x* axis for joint 2 = 22°.

After putting the measurement from Table 3 in Equations (1) and (2), Equations (3) and (4) are achieved. For the conditions sitting and standing, the mathematical model presents the information of the coordinate of the hip joint in Equations (3) and (4) as shown below.

$$a_e(\varnothing_1, \varnothing_2) = 0.4831 \tag{3}$$

$$b_e(\varnothing_1, \varnothing_2) = -1.9466 \tag{4}$$

The hip joint coordinate is (0.4831, −1.9466) as per Equations (3) and (4). This mathematical analysis regarding the motions of the knee and ankle joints at the sitting condition and the resultant condition is the necessary output. The outcome is achieved from the total derivatives of the earlier-mentioned kinematic Equations (1) and (2):

$(a_e, b_e)$ is the variable of both $(\varnothing_1, \varnothing_2)$. Therefore, the incorporation of two partial derivatives are utilized in the total process of calculation. The vector form of the position coordinate and angular variations is shown in Equation (5).

Using the method described in [12], $J$ is obtained as a 2 × 2 matrix as given below.

$$J = \begin{bmatrix} -33.5186 & -17.7326 \\ 0.4831 & 3.0909 \end{bmatrix} \tag{5}$$

where $J$ = linear velocity of the system.

### 3.2. The Standing Condition as Case II

The acquired measurements of the designed lower limb were $x_1 = 16$ inches, $x_1 = 18$ inches; $\varnothing_1 = 87°$; and $\varnothing_2 = 89°$ for standing.

Similarly, $J$ (linear velocity) [12] is a $2 \times 2$ matrix as shown in Equation (6), given by

$$J = \begin{bmatrix} -11.8755 & -1.2735 \\ 27.0708 & 17.9548 \end{bmatrix} \tag{6}$$

The matrix gives mathematical expression of the partial derivatives of the functions $a_e(\varnothing_1, \varnothing_2), b_e(\varnothing_1, \varnothing_2)$ related to the joint displacements regarding angular changes as $(\varnothing_1, \varnothing_2)$ in Equations (5) and (6). The matrix J presents the Jacobian Matrix.

### 3.3. Characteristic Equation Generation of Designed Prosthetic Limb from Jacobian Matrix

The obtained characteristic polynomial for case I can be obtained using the matrix method described in [12] from the Jacobian Matrix as given in Equation (5).

The achieved polynomial is as given below [12]:

$$f_1(t) = t^2 + 30.4277t + 112.1692 \tag{7}$$

In case II, in a similar way, the characteristic polynomial can be obtained using the Jacobian matrix given in Equation (6):

The achieved polynomial is as given below

$$f_2(t) = t^2 - 29.8303t + 178.7476 \tag{8}$$

The achieved characteristic equations of the designed system show the nonlinear presentation of the designed artificial leg.

The closed loop transfer function of the designed prosthetic limb is given below

$$G_{CLP} = \frac{9.904e27s^2 + 3.013e29s + 1.111e30}{1.981e28s^2 - 5.916e27s + 2.881e30} \tag{9}$$

The closed loop transfer function of the normal limb as developed by the authors is given below [12]

$$G_{CLN} = \frac{2.815e14s^2 + 7.216e15s + 2.602e16}{5.629e14s^2 - 1.256e16s + 3.109e16} \tag{10}$$

### 3.4. Lyapunov Stability Analysis of Designed Prosthetic Lower Limb

The Lyapunov stability [20] analysis is proposed for the nonlinear system stability analysis of the designed prosthetic limb, as mentioned in Equation (9).

$$\text{Since } e(s)G_c(s) \cdot G_p(s) = Y(s) \tag{11}$$

and $r(s) - y(s) = e(s)$, and assuming $r(s) = 0$, it can be stated that $y(s) = -e(s)$.

In this instance, $r$(s) = reference input, $y$(s) = system output and $e$(s) = error value, $G_c(s)$ = controller transfer function and $G_p(s)$ = process transfer function. Now, the mathematical model is shown in Equation (12) according to Equation (11).

$$e(s)\left(9.904e27s^2 + 3.013e29s + 1.111e30\right) = -e(s)\left(1.981e28s^2 - 5.916e27s + 2.881e30\right) \tag{12}$$

Using Inverse Laplace Transform, the transfer function is shown in Equation (13)

$$\left(9.904e27\ddot{e} + 3.013e29\dot{e} + 1.111e30e\right) = 0 \tag{13}$$

Now, taking the scalar positive definite function, which is given by

$$V(x) = \frac{1}{2}S_1 \cdot x_1^2 + \frac{1}{2}S_2 \cdot x_2^2 + \frac{1}{2}S_3 \cdot x_3^2 \qquad (14)$$

where $S_1 > 0, S_2 > 0$.

Now, taking the derivative with respect to time, t, yields

$$\dot{V}(x) = S_1 \cdot x_1 \cdot x_2 + x_2 \cdot x_3(S_2 - b_1 \cdot S_3) - x_3 \cdot (a_1 \cdot S_3 \cdot x_3 + c_1 \cdot S_3 \cdot x_1) \qquad (15)$$

For the positive definite function V, another positive definite function U is needed such that $\dot{V}(x) = -U(x)$. Now, the coefficients are taken in such a manner that $\dot{V}(x) = -U(x)$. Therefore, it is taken as

$$(S_2 - b_1 \cdot S_3) = 0 \qquad (16)$$

$$S_1 = 0 \qquad (17)$$

Now, substituting Equations (16) and (17) in Equation (15) and also $\dot{V}(x) = 0$, $V(x) \to \infty$ as $x \to \infty$ and $\dot{V}(x) = 0$ are obtained. A way of showing that $\dot{V}(x)$ being negative semi-definite is sufficient for asymptotic stability is to show that the *x* axis is not a trajectory of the system. For $\dot{x}_1 = x_2 = 0$ and $\dot{x}_2 = x_3 = 0$, it is shown that $x_1 = m$ (constant). The equilibrium state at the origin of the system is asymptotically stable. Therefore, the mentioned system in this work is asymptotically stable. The overall control law design and development of the efficient system for artificial lower limbs is the main focus of the work.

In Figure 7 from Equations (9) and (10) in Section 3.3 the output of the normal limb compared with that of the designed prosthetic limb is shown, where the normal limb system is mentioned as sys with a red line and the prosthetic limb system is mentioned as sys1 with a yellow line.

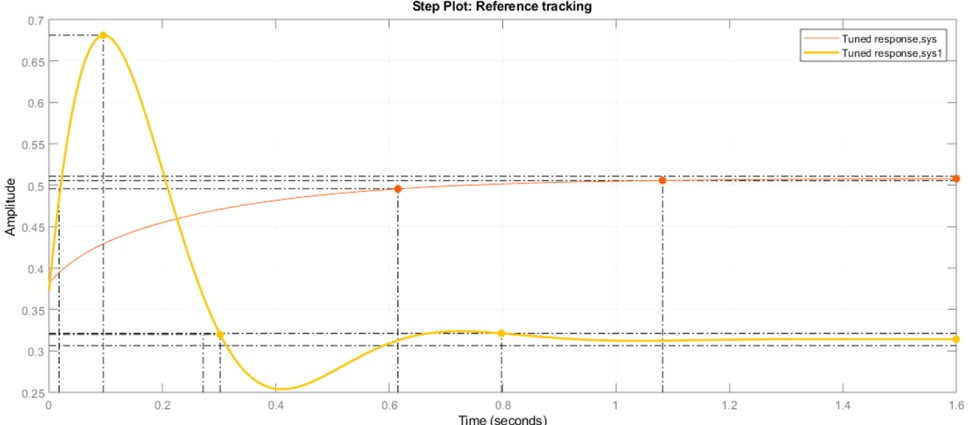

**Figure 7.** Graphical Presentation of Designed Normal Limb (sys) and Prosthetic Lower Limb (sys1) without tuning.

The system stability improvement was performed through a PIDF2 controller. In Figure 8, the output of the normal limb compared with that of the designed prosthetic limb, where the normal limb system is mentioned as sys with a red line and the prosthetic limb system is mentioned as sys1 with a yellow line.

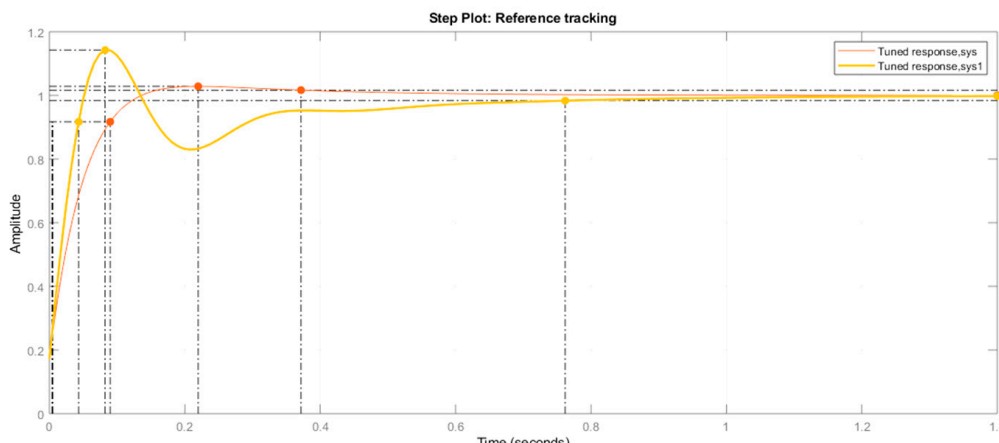

**Figure 8.** Graphical Presentation of Normal Limb (sys) and Designed Prosthetic Lower Limb (sys1), with PIDF2 (2 Degrees of Freedom type) Controller with tuning.

According to Table 4, the rise time has changed in the tuned condition of the designed prosthetic lower limb system. The settling time is decreased in the designed limb's dynamic characteristics, which indicates that after tuning stabilized output is obtained within a comparatively shorter period of time. The overshoot was decreased in the designed prosthetic limb where better stability was achieved, as a minimum time was needed to reach the required output. The peak value had a notable difference in between two mentioned systems such as the designed and the tuned systems. The change in the response time was recognizable after the PIDF2 type controller application on this 2DoF system. The proportional constant was lesser after tuning the designed system. This type of controller is suitable to use in this nonlinear system stability analysis as the system behavior is non-deterministic. After observing the graphical presentations and the characteristics values in Figures 7 and 8, the PIDF2 controller showed suitable nonlinear system stability with the dynamic characteristics of the designed prosthetic lower limb.

**Table 4.** Performance and Robustness Parameters of the Normal and the Designed Prosthetic Lower Limb using a PIDF2 Controller.

| Tuning Condition | Rise Time (Seconds) | Settling Time (Seconds) | Overshoot (%) | Peak | Proportional Constant ($K_p$) | Integral Constant ($T_i$) | Derivative Constant ($K_d$) | Response Time (Seconds) | Transient Behavior |
|---|---|---|---|---|---|---|---|---|---|
| Without tuning | 0.0302 | 0.798 | 117 | 0.681 | 1.1864 | NA | NA | 0.3515 | 0.6 |
| With tuning | 0.0393 | 0.762 | 14.2 | 1.14 | 0.41239 | 0.0084228 | 0 | 0.05571 | 0.6 |

## 4. Conclusions and Future Scopes

The locomotion control of the artificial lower limb movement of the gait cycle of a human being with the help of a kinesiological joint movement signal has been implemented in the present work. Gait synchronization of the prosthetic limb with sound lower limb movement has been performed. The Range of Motion (RoM) analysis for the artificial limb movement was made for the gait pattern implementation compared with the normal limb's motion features. The angular output of the designed prosthetic and normal knee joints differed in a range from 2 to 7 degrees and that of the ankle joints from 0 to 5 degrees. The voltage output of the built prosthetic and normal knee joints differed in a range from 0.1 to 0.3 volts and that of the ankle joints from 0.2 to 0.4 volts. The obstacle handling was incorporated with the artificial lower limb with body balancing attributes. The kinematic artificial system design and mathematical model generation is the most unexplored method utilized in this model design. The nonlinear stability analysis of the designed prosthetic limb was performed using a Lyapunov stability method suitable for system behavioral analysis, and

asymptotic stability was achieved. The use of PIDF2 controller improved the stability of the designed artificial limb system with proper adjustment of the tuning parameters.

A system design with better Degrees of Freedom is the future challenge. Analysis of leg types with different body weights and, according to that, servo motor movement analysis for robot-legs can be explored. The obstruction avoidance capability during motion can be added as special feature of this unilateral transtibial lower limb.

**Author Contributions:** Conceptualization, S.D. and B.N.; methodology, S.D. and D.N.; software, S.D.; validation, S.D.; formal analysis, S.D.; investigation, S.D.; resources, S.D.; data curation, S.D.; writing—original draft preparation, S.D.; writing—review and editing, S.D. and D.N; visualization, S.D.; supervision, D.N. and B.N.; project administration, B.N.; funding acquisition, S.D. All authors have read and agreed to the published version of the manuscript.

**Funding:** This research received no external funding.

**Institutional Review Board Statement:** Not applicable.

**Informed Consent Statement:** Not applicable.

**Data Availability Statement:** Not applicable.

**Acknowledgments:** The authors are thankful for the support of the Indian Institute of Information Technology Kalyani.

**Conflicts of Interest:** The authors declare no conflict of interest.

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
