# Peer review of "Design Analysis of Prosthetic Unilateral Transtibial Lower Limb with Gait Coordination"

_prosthesis, doi:10.3390/prosthesis5020040_

Round 1

Reviewer 1 Report

I find this work very interesting, however as a clinician, I would love to have more information presented about the applicability of this science to the real-life application in prosthetic design and implementation.

The manuscript is mostly free from errors in English language, however there are a few places in which there is some editing required.  For example, use of the word "Till" in place of "Until" in line 49.  The manuscript should be reviewed carefully for errors similar to this one, although it seems very readable to me.

Author Response

Please see the attachment regarding the modified paper and find the below-mentioned cover letter for the authors' responses to the respected reviewer's comments and suggestions.

Review Report of Submission Id – Prosthesis-2380160

Title of the Paper: Design Analysis of Prosthetic Unilateral Transtibial Lower Limb with Gait Coordination.

The authors of the paper entitled “Design Analysis of Prosthetic Unilateral Transtibial Lower Limb with Gait Coordination” are very much grateful to the respected reviewer for the valuable comments of the research paper Id Prosthesis-2380160 for the submission in the esteemed journal ‘Recent Advances in Foot Prosthesis and Orthosis’.

The comments of the respected reviewer are presented below with the authors’ responses to the valuable comments.

Review 1:

Comments and Suggestions for Authors:

I find this work very interesting, however as a clinician, I would love to have more information presented about the applicability of this science to the real-life application in prosthetic design and implementation.

Comments on the Quality of English Language:

The manuscript is mostly free from errors in English language, however there are a few places in which there is some editing required.  For example, use of the word "Till" in place of "Until" in line 49.  The manuscript should be reviewed carefully for errors similar to this one, although it seems very readable to me.

Authors’ responses: (All the changes are in track change mode and marked yellow)

  1. In the abstract of the paper some information is added related to the social impact of the prosthetic system design for the transtibial amputees which is very much essential for the improvements of the rehabilitation process.
  2. In the introduction of the paper, some information is added related to the applicability of the prosthetic system design and implementation in real life.
  3. In the section 2, the heading ‘Methodology’ is replaced by ’Materials and Methods’ as per instruction given in the revision mail. Some information is added in section 2.2 related to present the significance of real-time prosthetic system design.
  4. The names are added in the contribution section at the end of the paper as per instruction.
  5. The word ‘till’ is used in the mentioned line.

Reviewer 2 Report

1- I think that the title is to replace by 

Design Analysis of Prosthetic Unilateral Transtibial Lower 2 Limb with Gait Coordination

2- Abstract is need to improved according the basic logical  question (Why, How, What )

3- Introduction is need to supported with more modern references 

4-I think that the authors must show the method of fixation the Encoder at the knee and ankle joint  

5- This designed for permanent using ,but the Encoder is big !.How the authors solved this problem.

6- Results are needed to supported by more discussion 

7- Conclusions are need to rewritten to specific clear points

1- I think that the title is to replace by 

Design Analysis of Prosthetic Unilateral Transtibial Lower 2 Limb with Gait Coordination

2- Abstract is need to improved according the basic logical  question (Why, How, What )

3- Introduction is need to supported with more modern references 

4-I think that the authors must show the method of fixation the Encoder at the knee and ankle joint  

5- This designed for permanent using ,but the Encoder is big !.How the authors solved this problem.

6- Results are needed to supported by more discussion 

7- Conclusions are need to rewritten to specific clear points

Author Response

Please find the review report of Submission Id – Prosthesis-2380160

Review Report of Submission Id – Prosthesis-2380160

Title of the Paper: Design Analysis of Prosthetic Unilateral Transtibial Lower Limb with Gait Coordination.

The authors of the paper entitled as “Design Analysis of Prosthetic Unilateral Transtibial Lower Limb with Gait Coordination” are very much grateful to the respected reviewer for the valuable comments of the research paper Id Prosthesis-2380160 for the submission in the esteemed journal ‘Recent Advances in Foot Prosthesis and Orthosis’.

The comments of the respected reviewer is presented below with the authors’ responses to the valuable comments.

Review 2:

Comments and Suggestions for Authors:

1- I think that the title is to replace by 

Design Analysis of Prosthetic Unilateral Transtibial Lower 2 Limb with Gait Coordination

2- Abstract is need to improved according the basic logical  question (Why, How, What )

3- Introduction is need to supported with more modern references 

4-I think that the authors must show the method of fixation the Encoder at the knee and ankle joint  

5- This designed for permanent using ,but the Encoder is big !.How the authors solved this problem.

6- Results are needed to supported by more discussion 

7- Conclusions are need to rewritten to specific clear points

Comments on the Quality of English Language:

1- I think that the title is to replace by 

Design Analysis of Prosthetic Unilateral Transtibial Lower 2 Limb with Gait Coordination

2- Abstract is need to improved according the basic logical  question (Why, How, What )

3- Introduction is need to supported with more modern references 

4-I think that the authors must show the method of fixation the Encoder at the knee and ankle joint  

5- This designed for permanent using ,but the Encoder is big !.How the authors solved this problem.

6- Results are needed to supported by more discussion 

7- Conclusions are need to rewritten to specific clear points

Authors’ responses: (All the changes are in track change mode and marked with blue)

  1. The title of the paper is given as per suggestion.
  2. In the abstract of the paper, more description is added to give information about the Why, How, What of the prosthetic limb design.
  3. More recent references are added in the reference section and explained the work the Introduction section.
  4. The fixation of the rotary encoders are shown through captured pictorial presentations in Fig. 2 during the model preparation. It is explained in the section 2.1.
  5. As the encoder is big in size, the attachment process and the reason to use the material for the kinematic joint motion detection is explained in the 2.1 section.
  6. The result section is explained with more discussions and explanations to give better understanding of the prosthetic limb design and comparison with the normal limb walking style to show the body balancing feature of the amputees.
  7. The conclusion part is described with more detailed explanation to present the productive outcome and significance to utilize the nonlinear system stability analysis with mathematical model generation while conducting the system building.

Reviewer 3 Report

I would like to thank the authors for the interesting research.

However, I have some important comments.

The purpose of the work is not formulated in the introduction. The formulation of the research aim is essential.

The authors discuss the novelty of the conducted research (line 73) in the methodology. I think the novelty of the work, should be disclosed in the introduction and discussed in the discussion.

The authors also include discussion elements in the methodology (lines 100-103). I think it is necessary to move them into the discussion part.

Since the goal is not formulated in the manuscript, I cannot evaluate the methodology. The process of this research remains unclear.

After formulating the goal, it is necessary to specify the methodology.

The authors are missing the discussion. I think the discussion is necessary. It is necessary to reveal the authors' contribution to existing Prosthetic Unilateral Transtibial Lower Limb with Gait Coordination solutions.

I suggest adjusting the research manuscript in principle.

Sincerely.

Author Response

Please find the Review report of Prosthesis-2380160.

Review Report of Submission Id – Prosthesis-2380160

Title of the Paper: Design Analysis of Prosthetic Unilateral Transtibial Lower Limb with Gait Coordination.

The authors of the paper entitled as “Design Analysis of Prosthetic Unilateral Transtibial Lower Limb with Gait Coordination” are very much grateful to the respected reviewer for the valuable comments of the research paper Id Prosthesis-2380160 for the submission in the esteemed journal ‘Recent Advances in Foot Prosthesis and Orthosis’.

The comments of the respected reviewer is presented below with the authors’ responses to the valuable comments.

Review 3:

Comments and Suggestions for Authors:

I would like to thank the authors for the interesting research.

However, I have some important comments.

The purpose of the work is not formulated in the introduction. The formulation of the research aim is essential.

The authors discuss the novelty of the conducted research (line 73) in the methodology. I think the novelty of the work, should be disclosed in the introduction and discussed in the discussion.

The authors also include discussion elements in the methodology (lines 100-103). I think it is necessary to move them into the discussion part.

Since the goal is not formulated in the manuscript, I cannot evaluate the methodology. The process of this research remains unclear.

After formulating the goal, it is necessary to specify the methodology.

The authors are missing the discussion. I think the discussion is necessary. It is necessary to reveal the authors' contribution to existing Prosthetic Unilateral Transtibial Lower Limb with Gait Coordination solutions.

I suggest adjusting the research manuscript in principle.

Authors’ responses: (All the changes are in track change mode and marked with green)

  1. The purpose of the work is formulated in the ‘Introduction’ section as per suggestion.
  2. The discussion regarding the novelty of the conducted research is disclosed in the ‘Introduction’ section and discussed in the ‘Results and Discussions’ section under discussion part.
  3. The discussion elements in the methodology (‘Materials and Methods’ section) is moved to the discussion part under the ‘Results and Discussions’ section as per suggestion.
  4. The goal is formulated in the manuscript now clearly in the ‘Introduction’ and ‘Materials and Methods’ sections to understand the need of the research work and the presented solution for the amputees to achieve advanced gait rehabilitation.
  5. After formulating the goal in the ‘Introduction’ section, it is specified in the ‘Materials and Methods’ section as per suggestion.
  6. The authors' contribution to existing Prosthetic Unilateral Transtibial Lower Limb with Gait Coordination solutions is mentioned under the ‘Authors’ contribution’ section sequentially and the discussions are given in the ‘Results and Discussions’ section to understand the different points of the contribution to the research work and the solution for the lower extremity design for the transtibial amputees.
  7. The manuscript of the research is adjusted in principle as per suggestions.

Round 2

Reviewer 2 Report

The last version after revision notes is suitable for  good work .I think that this version is deserve publishing